# A Recognition Method for Soft Objects Based on the Fusion of Vision and Haptics

**DOI:** 10.3390/biomimetics8010086

**Published:** 2023-02-20

**Authors:** Teng Sun, Zhe Zhang, Zhonghua Miao, Wen Zhang

**Affiliations:** Intelligent Equipment and Robotics Lab, Department of Automation, School of Mechatronic Engineering and Automation, Shanghai University, Shangda Street No. 99, Baoshan District, Shanghai 200444, China

**Keywords:** deep network, haptic exploration, graph neural network, sensor fusion, object perception

## Abstract

For humans and animals to recognise an object, the integration of multiple sensing methods is essential when one sensing modality is only able to acquire limited information. Among the many sensing modalities, vision has been intensively studied and proven to have superior performance for many problems. Nevertheless, there are many problems which are difficult to solve by solitary vision, such as in a dark environment or for objects with a similar outlook but different inclusions. Haptic sensing is another commonly used means of perception, which can provide local contact information and physical features that are difficult to obtain by vision. Therefore, the fusion of vision and touch is beneficial to improve the robustness of object perception. To address this, an end-to-end visual–haptic fusion perceptual method has been proposed. In particular, the YOLO deep network is used to extract vision features, while haptic explorations are used to extract haptic features. Then, visual and haptic features are aggregated using a graph convolutional network, and the object is recognised based on a multi-layer perceptron. Experimental results show that the proposed method excels in distinguishing soft objects that have similar appearance but varied interior fillers, comparing a simple convolutional network and a Bayesian filter. The resultant average recognition accuracy was improved to 0.95 from vision only (mAP is 0.502). Moreover, the extracted physical features could be further used for manipulation tasks targeting soft objects.

## 1. Introduction

It is a common sense that humans instinctively combine multi-modal sensory data such as vision, touch, audition, etc., to perceive the surrounding environment. While some of the sensing modalities are non-contact such as vision and audition, others require interactive action for perception, such as touch [1]. More narrowly, visual feedback obtains global scene information containing semantic and geometric object properties, which can be used for accurate object reaching, while haptic feedback obtains current contact conditions through haptic interactions locally, which can be used for accurate localization and manipulation control [2]. For object perception and dexterous manipulation tasks, these two types of sensing modalities are essential and complementary. In effect, the notion that the interdependence and concurrency of vision and touch aid perception and manipulation has been proved in neuroscience [3]. Despite the vast advancements in processing and learning visual data and the huge research interest in evolving the artificial sense of touch, the optimal integration of visual and haptic information has not yet been achieved.

To solve this dilemma, technologists obtained inspiration from the visual–haptic contribution in human sensorial loops in accordance with the contour following strategy that humans use for object recognition. From a psychophysics and neuroscience side, many researchers have been trying to explain how tactile and visual information contribute in order for humans to interpret their environment. Klatzky et al. suggested that both vision and touch rely on shape information for object recognition [4]. They also studied different exploratory procedures that humans apply for tactile object recognition [5] and revealed the superiority of tactile perception in the presence of vision [6]. Demarais et al. studied the performance of visual, tactile, and bimodal exploration of objects for both learning and testing procedures for object identification [7]. Calandra et al. proposed an end-to-end action conditional model that learns re-grasping policies from raw visual–tactile data [8]. Gao et al. designed deep architectures for haptic classification using both haptic and visual signals [9]. Lee et al. proposed a novel framework for cross-modal sensory data generation for visual and tactile perceptions [10]. Yu et al. proposed a framework that fuses tactile and visual sensing to recover the pose and contact formation of an object relative to its environment, for robotic packaging [11]. Fazeli et al. proposed a methodology based on the hierarchical Bayesian model to emulate multisensory fusion in a robot that learnt to play Jenga [12]. These studies tried to integrate visual and haptic data for object recognition, material classification, manipulation, or grasp control.

Many other works focusing on the integration of vision and touch have been presented, among which, some are studied for better object perception, such as [13,14,15], and others for precise manipulation control, such as [16,17,18]. For objects which have the same appearance, but different interior fillers or made from different materials, they become difficult to identify with vision solely, and if they can deform during interaction, it will become more challenging. Most of the studies rely on haptics to deal with soft objects. Gemici and Saxena tried to model and learn the physical properties of deformable food objects through actions that can extract haptic data. They used the learnt properties to plan appropriate manipulation actions [19]. This also demonstrates the significance of obtaining heterogeneous features of the objects through different exploratory actions, in turn, the features can serve as guidance for planning appropriate control strategies. Furthermore, Yim et al. proposed a unified data-driven framework for modelling and rendering the stiffness and friction of a soft object with haptic feedback [20]. Other works are shown in [21,22].

As can be seen from the preceding information, visual and haptic integration is vital and has attracted the interest of an increasing number of researchers with emerging, related studies. Nevertheless, in the majority of previous studies, for the fusion task, usually a deep network was used for vision and a simple classifier for haptics, and only the respective results were fused with simple convolutional networks or machine learning methods. The vision and haptic features were not integrated properly. This is because traditional neural networks typically process Euclidean structured data (such as image data), whereas haptic data often have low dimensions and are difficult to represent in an n-dimensional linear space, thus making traditional neural networks unsuitable for extracting haptic features. The non-Euclidean haptic data can be viewed as graphs, and the graph neural network (GNN) is an emerging method for dealing with such data, as proposed in [23,24].

Based on graph convolution neural networks (GCN), an end-to-end visual–haptic fusion perception method has been proposed. Through which, the vision- and haptic-related features were properly integrated for improved object recognition and adaptive control. Specifically, haptic sensing provides more informative features that are difficult to obtain by vision alone; meanwhile, vision provides not only the initial non-contact perception to narrow down the identity scope of the object, but also the accurate relative location to reach for further haptic interaction. The proposed GCN model integrates both the vision and haptic features, and the object is recognised through a multilayer perceptron (MLP). The experiment results show that the perception accuracy increased from 0.58 (vision only) to 0.95 with the fusion of vision and touch. Moreover, a Conv1D network and Bayesian filer were used as comparisons and the results were improved with the proposed method. Additionally, it was proven that control can be integrated with perception, which means that the action controlled (haptic exploration) is used for better perception, while the better perception will, in turn, facilitate the control to be more adaptive.

Regarding the contribution: first, a method for integrating vision and haptic features was designed, which showed higher perception accuracy when dealing with soft objects; second, a software system was provided that can control the robot for autonomous vision perception and haptic explorations; third, this method could be easily extended with more sensing modalities such as audition, smell, etc. Finally, this work could lead to more complex tasks such as adaptive force or grasp control [25].

This paper is structured as follows: In Section 1, the background and contributions of this work are shown briefly. Then, the proposed algorithms for handling the perception tasks via the fusion of vision and haptics is represented in Section 2. After that, the experimental validations using an assembled robot platform are expressed, and the results are analysed and discussed in Section 3. Finally, a short conclusion and future work are provided in Section 4.

## 2. Materials and Methods

The goal of this work is to recognize objects that are difficult to detect based only on vision. To target this challenge, an end-to-end visual–haptic fusion method for object perception was proposed. The overall control flow is expressed in Figure 1. The first part is the multimodal feature extraction part, in which respective sensors are driven by the robot platform, and visual and haptic features are extracted through the YOLOv5 network and haptic explorations, respectively (Section 2.1 and Section 2.2). In more detail, the YOLOv5 network is used for the initial perception of the object and to locate its position for the haptic sensor to reach. Then, as the key sensor fusion part, the multi-modal features are concatenated through a GCN model (Section 2.3). In the object-classification part, the features integrated by the GCN model are transformed into the three-layer MLP. The feature dimension is reduced to 128 after the first two layers, and after the last layer, the feature dimension is reduced to (*M*, 1). After that, a layer of soft-max is connected, and the object identity is decided, which has the highest probability (the total number of object categories is *M* = 10 in this study; more details can be found in Section 3.2).

### 2.1. Vision Perception Model

Using a depth camera, the RGB data containing rich colour information and the depth data storing the distance information can be obtained. In this work, the RGB colour data were mainly used for the initial perception and localization of the target object, while the depth data provided the required distance for the robot arm to approach the object and initiate haptic exploration. To better combine these two kinds of information, spatial alignment was used to attach the corresponding depth value to each colour pixel. Moreover, to reduce the effects brought about by different lighting conditions, which may downgrade the perception accuracy, image preprocessing was carried out. As the main step, the adaptive histogram equalization was applied on the **v** (value) element after converting the original RGB data to HSV gamut in order to lower the brightness variation of the object and environment.

To detect the object with visual feedback, the YOLOv5 network was selected as the perception model based on its real-time performance and a lack of dependence upon the sample size. It originated from the YOLOv1–YOLOv4 models, which were first introduced by [26] and have been widely used for object recognition. As described in [26], for YOLOv5, the data were first input to CSPDarknet for feature extraction, and then fed to PANet for feature fusion. Finally, the detection results were output (object class, confidence score, bounding box). In this work, the modal was trained, starting from the pre-trained YOLOv5l modal based on the COCO dataset.

### 2.2. Haptic Explorations

After the recognition trial through the YOLOv5 network was completed, if the recognition result was incorrect or the confidence score was not satisfied, the haptic features would be obtained through haptic explorations, and object recognition by processing the integrated vision and haptic features would be carried out.

#### 2.2.1. Primary Work

To locate the object and guide the robot to approach its surface (or margin) for haptic interaction, a method was proposed, which is based on the bounding box obtained from the YOLOv5 detection result. The detailed process is introduced in the following section.

If the YOLOv5 network returned detection results, first, the centre of the box would be found as the initial centroid of the object. The box contained a large area that did not belong to the object, therefore, it was extracted as the region of interest (ROI), and the Canny edge detection was used to find the contour within. Then, the centroid of the contour was calculated as the centre of the object (red dot in Figure 1). After that, with the spatial alignment method, the corresponding depth of the centroid (*z*) was obtained. At last, the coordinates of the centroid was transformed from the image frame to the world frame, and the location for the robot arm to reach the object was acquired ($P_{c}$).

It is not necessary to start haptic exploration from the centroid of the object, therefore, the starting point could also been changed by adding offset based on the size of the object. However, this method requires the object to fully appear in the image; otherwise, the provided centroid is not the real centroid of the object. Thus, in this work, the reaching point was fixed to the top middle of the object by adding an *W*/2 offset from the centroid of the object, where *W* is the width of the object.

After reaching the surface of the object, the robot will be controlled to carry out haptic explorations to extract the corresponding features, for which contact state estimation is an essential procedure. This is because the current contact location and forces are essential for determining the next exploration location and interpreting the haptic features at each step. There are two main frames, the arm base {O} and hemisphere tip frame {E} (which is attached to the end-effector). In this study, control was based in {O}, while the contact state estimation was based in {E}. The coordinates of the contact point were set to $P_{c}^{E}={\left[x_{e},y_{e},z_{e}\right]}^{T}$ in frame {E}. Based on the equilibrium equation in [27], $P_{c}^{E}$ was calculated and $P_{c}^{O}$ in {O} was obtained through coordinate transformation. It should be noted that the contact tip used in this work was made by 3D printing and is rigid. Soft tips that can deform during contacting will be studied in future work.

#### 2.2.2. Robot Arm Control and Exploration Procedures

As the primary mobile component, the robot arm is controlled in order to carry out two main actions—the first is to approach the object with the visual feedback and the second is to implement haptic explorations such as compression, surface following, or contour following based on the haptic feedback.

After the robot approached the object and the contact position and forces were obtained, the next position to be explored was derived. As proposed in [28], a geometrical method was used, by which, a force $F_{N}^{\prime }$ was used to make sure the contact normal force is under control and a virtual force $F_{T}^{\prime }$ is used to ensure that the robot arm moves forward following the surface of the object. Then, the direction of a guiding force $F_{G}$ was obtained based on the correcting force $F_{N}^{\prime }$ and current contact state, for the surface or edge following actions. The next exploration location $P_{next}$ was decided upon with a velocity parameter, which can decide the exploration speed. After the next exploration location was obtained, the control signals for each joint were calculated following the inverse kinematics of the arm.

For the compression action control, the robot arm is required to move along the normal direction of the object surface. For this task, the contact tip was controlled to move along the normal force direction for a distance *L*. During the process, the forces at the first contact ($f_{1}$) moment and after it moved *L* ($f_{2}$) were recorded and the stiffness was calculated by $\frac{f_{2}-f_{1}}{L}$ at this location. To increase the reliability of the obtained stiffness, it was calculated repeatedly at five places and the average was computed. For the contour following, the process was similar to the surface following the action—the only change required was to locate the contact tip at the edge of the object.

These exploration procedures were carried out to extract the physical properties of the contacting object as supplementary to visual features. Specifically, the surface friction and roughness were obtained through the surface following, while the contour following action gave insights into the external surface curvature, which can also be used for 3D reconstruction. Furthermore, since the objects used in this work were soft, the compression action was required to obtain the average stiffness.

### 2.3. Features Fusion with GCN

With the vision-related and haptic-related features extracted, a method that can integrate the multi-modal features was required for an improved object recognition performance. Traditional neural networks generally process Euclidean structured data, but there are a lot of data without rules in real life, such as topology, knowledge graphs and so on, as well as the haptic features obtained. They are not translation invariant, so traditional CNNs or RNNs are not suitable for processing this type of data.

As the research continued, graph neural networks (GNNs) have been proposed to deal with non-image data or so-called graphs. A graph represents the relationships (edges) between a set of entities (nodes) and has three basic attributes, node attributes, edge attributes, and global attributes, as shown in Figure 2. Xiao et al. summarized and generalized the existing models of graph neural networks to provide a generic structure [23]. Seenivasan et al. tried to use a graph network for surgical scene understanding [29].

Since the experimentally obtained haptic data often have characteristics such as low dimensionality and data irregularity, GNNs can be used to analyse such data. Many related methods have been proposed in recent years. The following methods are available: graph generative adversarial networks (GAN) [30], graph auto-encoders (GAE) [31], graph attention networks (GAT) [24], and graph convolution neural networks (GCN). In this study, the GCN method was selected, which is a cleverly designed method to extract features from graph data, and can then be used to perform node classification, graph classification, and link prediction on the graph data.

Thus, a graph convolution model was built, and specifically, the network has twelve layers. The first layer uses GCNConv convolution, the second layer uses BatchNorm for the normalization function, the third layer uses ReLU for the activation function, and the fourth layer uses TopKPooling for the downsampling function. These four layers are combined together as one block, with a total of three blocks. A general overview of the network is shown in Figure 3.

For the integration task based on the GCN, the obtained visual features through the deep network and the interpreted haptic features need to be modelled to a graph as G=(N,E), among them $N=\left\{n_{i}\right\}$, are the nodes representing the features vector incorporating all the multi-modal features ($n_{i}=\left[T,V\right]$), $E=\left\{e_{ij}\right\}$, and $e_{ij}$ is the edge weight between the nodes.

For this purpose, first, the data were projected into Euclidean space and the Euclidean distance between every two nodes was calculated. The calculated Euclidean distance was fed into the sigmoid non-linear activation function (which presents the value of the Euclidean distance between [0–1]) and its output was used as the edge weights between the nodes. The vector representation of the edges is shown in Equation (Equation 1).
(1)$$E=Concat(\sum_{n=1}^{n=C_{r}^{2}}\sigma \left(\sqrt{{\left(x_{2}-x_{1}\right)}^{2}+{\left(y_{2}-y_{1}\right)}^{2}}\right))$$
where *r* represents the total number of nodes, $C_{r}^{2}$ represents the number of combinations between r nodes, σ represents the sigmoid non-linear activation function, $\left(x_{2},x_{1}\right)$ and $\left(y_{2},y_{1}\right)$ represents the *x* and *y* coordinates of two nodes projected into a two-dimensional linear space, respectively. The Concat operation represents the stitching together of nodes and edge weights between nodes to obtain a $C_{r}^{2}$-dimensional vector.

The global attributes (*g*) are determined by a vector of points (*N*) and a vector of edges (*E*) together. A learnable parameter λ is assumed to represent the weight of the point vectors and 1−λ is the weight of the edge vectors. At first, λ = 0.5 is assumed, and the value of λ changes with the number of training iterations. The global vector *g* is defined as
(2)g=(λ)N+(1−λ)E

After defining the point features, edge features and global features, the entire graph is encoded, and the encoded features are shown in Equation (Equation 3).
(3)$$\begin{array}{cc} \; & G^{\prime }=E\left\|N\right\|g\\ \; & G=MLP(G^{\prime }) \end{array}$$
where $G^{\prime }\in {ℜ}^{\left(g+E+N\right)}$ is the concatenated representation of *g*, *E*, and *N*; $MLP\in {ℜ}^{128\times (g+E+N)}$ is one linear transformation which embeds the spatial relation information (g+E+N) into a 128-dimensional representation, and transforms $G^{\prime }$ into the scalar *G*.

After the GCN modal has been decided, the edge weights connecting the nodes need to be trained and learnt. For this purpose, the features $\left\{n_{i}\right\}$ of the proposed spatial modality graph are refined iteratively *l* times as follows:(4)$$n_{i}^{\left(l+1\right)}=n_{i}^{\left(l\right)}+\sigma \left(W^{\left(l\right)}\left(\sum_{i\neq j}\alpha_{ij}^{\left(l\right)}e_{ij}^{\left(l\right)}\right)\right)$$
where $n_{i}^{\left(l+1\right)}\in {ℜ}^{D}$ indicates the feature of the ith graph node at time step *l*, and $\alpha_{ij}^{\left(l\right)}$ is the learnable normalized edge weight between node i and j at time step *l*, which is given by:(5)$$\alpha_{ij}^{l}=\frac{exp(e_{ij})}{\sum_{k\neq j}\left(e_{ij}\right)}$$

The weights $\alpha_{ij}$ will be decided through training. Formally, the loss is defined using the cross entropy (CE) loss, during the learning process. The proposed graph *G* changes dynamically during reasoning from one iteration to another, to make it easier to understand, Figure 4 is shown. The final output $n^{l}$ of the iterative reasoning module is fed to the classification module to classify *M* candidate objects (shown in Figure 1).

## 3. Experimental Validations

The proposed method has been experimentally tested using an integrated robot platform. To better verify its advantages, deceptive and relatively complex tasks were designed.

### 3.1. Robot Platform and Task Description

An experimental robot platform was assembled, which mainly consists of an Aubo i3 robot arm with a 6-axis force-torque sensor attached, a 3D printed hemisphere contact tip, a Realsense D435i RGBD camera, and different objects for validation. The software system was built based on ROS (www.ros.org (accessed on 23 December 2022)) in the Ubuntu system. The PC communicates with the force-torque sensor through the Ethernet, with a 200 Hz refresh rate under ROS. Meanwhile, to deal with soft objects, which can deform during haptic interaction, real-time force-torque feedbacks are essential. The feedbacks are further used to calculate contact locations and forces, which are used for position and velocity control of the robot arm. Specifically, the tactile sensing toolbox (TST) introduced in [32] was used for obtaining the necessary data at 100 Hz.

Regarding the tasks, the method was proposed to recognise the objects at satisfied accuracy with multiple sensing modalities. When detection results (such as confidence) that are obtained through the deep learning method using solely visual feedback are not satisfactory, additional perception strategies with extra features are required. New actions that extract features which belong to another sensing modalities will be executed. For example, when different objects have similar appearances, but have varied fillers, or are made from different materials, classifying the objects with only visual feedback is likely to fail. In such situations, features such as stiffness or surface friction will be more useful for recognising the object. For this purpose, the robot will be guided by vision to reach the object and carry out haptic explorations to extract the required features such as stiffness, surface friction, etc. With the proposed fusion method based on GCN, the visual features and haptic features are incorporated for improved object recognition. Moreover, to evaluate the advantages of the proposed integration method, a Conv1D method and a Bayesian filter were used as the comparisons during the experiments.

### 3.2. Objects Creation and Vision Model Training

For recognition tasks, different sets of objects were created. Moreover, objects with a similar surface and appearance but different interior materials (such as organs) were used to make visual perception more difficult. Meanwhile, the objects were soft and able to deform when force was applied during haptic interactions. As can be seen in Figure 5, all the objects are soft, for objects 0 and 6–9, the same soft cube was used with different fillers such as plasticine, silicone, and sponge.

The visual data of the objects were obtained through the D435i camera and labelled manually. In total, 1000 pictures were used for training and 600 for validating. After training the data for 500 epochs, the F1, recall–precision curves, and the mAP of each object were obtained. As shown in Figure 6, for objects such as the cuboid, paper box, or cylinder, which are unique, the average precisions are very high, with the highest being 0.989. However, for the created cubes that have a similar appearance, the average precision is only 0.502, which is not satisfactory for practical usage.

### 3.3. Implementation Procedures

After the vision perception modal was obtained, to verify the effectiveness of the proposed fusion method, a number of controlled experiments were carried out. The following steps were taken to implement the experiments, and Figure 1 can be referred for more details.

1.Initialise all the required variables. Check whether the robot has returned to the “work position”; if no, return it to that position. If yes, move it to the “vision position” and detect the object using the RGB data through the obtained YOLOv5 model, and record the results (confidence score, boxes and object identity).2.Check if the maximum confidence exceeds the pre-set threshold; if yes, output the corresponding object identity. If no, detect the reaching position of the object with the RGBD data and calculate the control angles for the robot arm joints.3.Control the robot arm to reach the object surface (“touch position”). Implement the compression action at five random locations and record the force changes. Then, return the robot to the top middle of the object to carry out the surface following within the object surface and record the contact forces and locations. After that, calculate the average stiffness and surface friction (mean and standard deviation).4.Normalise the collected features to fix the data between [0,1]. Carry out the data embedding with the normalised data and feed the embedded data into the GCN network, which is then connected with a three-layer MLP classifier. Using the classifier to process the integrated features, output the object identity with the highest probability. More details will be introduced in the following sections.5.Check whether the new confidence is satisfied. If “yes”, inform that the object has been detected successfully and print out the result. If “no”, return to the first step and repeat the perception process. In either case, the robot arm must return to the “work position”. Please check the Appendix A for more details about the experiemnts.

#### 3.3.1. Data Processing

The vision perception model was introduced in Section 2.1 and the results including the confidence score, bounding boxes, and object class were obtained. The haptic explorations were introduced in Section 2.2, which extracted the stiffness, surface friction, and roughness of the object. To visualise the haptic features, and to prove that they are not regular Euclidean data, 500 samples for each feature of object no. 6–9 were recorded and plotted in Figure 7. Then, the vision and haptic fusion method based on the GCN was used to integrate the multi-modal features before the data were required to be processed.

After all the features were obtained, a data matrix *D* with a size of S∗6 (*S* is the number of all samples) was constructed, in which the attributes in the first five columns included confidence, bounding box, object stiffness, surface friction and roughness, and the attribute in the sixth column was the object identity (ID). For a better view, examples of the features values are shown in Table 1. After that, the features values were required to be normalised to map the original data to a normal distribution with a mean of 0 and a standard deviation of 1.

To explore the optimum number of nodes in this model, experiments were carried out. The recognition results when using different number of features are shown in Table 2. During the experiments, first, the visual features were all used; then, one haptic feature was fused; after that, two haptic features were fused; and finally, all the three haptic features were fused. As shown in the table, with the fusion of haptic features, the accuracy rate increased. On the other hand, when all the haptic features were used, first, one visual feature was added, and then, two visual features were fused. As a result, when all the visual and haptic features were used, the recognition accuracy rate was the highest (up to 0.95).

As mentioned in Section 2.3, the features were integrated using a GCN modal G=(N,E); among them, *N* was the features vector of the five nodes representing both the vision and haptic features. To train the GCN modal, the five obtained features were extracted from $D(S_{t},\left(1^{\sim }5\right))$ as learning samples; $D(S_{t},6)$ as the ground truth object label, $S_{t}$ represents the data number for training.

In this paper, the PyTorch deep learning tool was used to assist in the training of both the YOLOv5 modal and the GCN model, and the NVIDIA GeForce RTX 3080Ti graphics card was used for GPU acceleration during training. For training the GCN model, the Adam optimizer was used, the batch size was set to 32, the learning rate was set to 0.002, the rest of the parameters were set to default, and the number of epochs was set to 50. The training and test loss of the model are shown in Figure 8.

#### 3.3.2. Object Recognition Tests

In order to verify the robustness and advantage of the proposed method, object recognition tests were carried out with a simple convolutional network (Conv1D) and a Bayesian filter for comparisons.

First, a control experiment was designed, in which only the network structure was modified from the graph neural network to a three-layer Conv1D network, while the other parameters were unchanged. The three-layer Conv1D network parameters are [[32, 64, 3, 2], [64, 128, 3, 2] and [128, 32, 2, 1]] respectively. After setting up the network architecture, the same configuration as the GCN modal was used for training, and the number of epochs for training was set to 50. After training, the analysis results were plotted and are shown in Figure 9. As shown in the figure, the average accuracy increased from 0.502 (vision) to 0.85 with the Conv1D method, and 0.95 with the proposed GCN modal using 50 testing samples for each object.

Moreover, a Bayesian filter, and KNN and SVM methods were utilised as the control methods, which were trained using the same training samples. With the same testing samples, the resultant recognition accuracies were 0.83, 0.81, and 0.82, respectively, which were similar with the Conv1D method (detailed experimental results are shown in Table 3). This also proves that the proposed method had a better performance. However, the Bayesian method has a better real-time performance and can be extended to an improved interaction method with an information-gain approach in future work.

Additionally, with the feedback from both vision and haptics, it is possible to acquire extra information of the object. For object-*, which has a metal object inside, its stiffness map was captured through the help of a vision guide and compression action. As shown in Figure 10, grids were drawn to make it easier to see the location of the metal object. In practise, the robot follows the visual guide to reach the initial position, and during the procedure, it relies on it to correct the location error. It can be seen that the stiff object inside the object was located, which proves the potential of using haptic exploration to locate interior objects with a distinctive stiffness that cannot be seen directly.

However, there were several wrong detections for the objects 0 and 6–9. They were considered to be caused by the unsteadiness of the robot arm and the force-torque sensor during the movement and data collecting procedures. Furthermore, more features will be added to improve the robustness of the model and increase recognition accuracy and universality for more objects. This can be carried out by adding more haptic exploration actions and features extracted with other sensing modalities.

### 3.4. Perception for Adaptive Force Control

This recognition method based on the fusion of vision and haptics can help the robot to classify objects more precisely and obtain additional physical information, such as stiffness or friction. In practise, this information can be further used to realise adaptive force control. For instance, the information can help the robot to decide the correct holding force when using a soft gripper with controllable grip force (like the hand shown in [33]), which will help to avoid damage to the object while avoiding slippage. To verify this, a corresponding experiment has been carried out.

As shown in Figure 11, the soft cubic box was used as the object and a DH-AG95 gripper was used to hold the box. The gripping force is controllable using the gripper with an impedance control method, and the force applied is divided into 20% to 100% of the maximum permissible force. The degree of closure of the gripper can also be controlled. The maximum permissible force has been calculated to be 30 N (on one side) when the closure degree is 30% of the maximum stroke (95 mm).

As mentioned in Section 3.2, the mean stiffness of the box was obtained to be 1.92 N/mm. Therefore, when the holding force is set to 20% (6 N) and the degree of closure is fixed to 30% (28.5 mm in total and 14.25 on one side) of the maximum stroke, the deformation of the object should be close to 3mm (due to the hardware limit, the smallest force is 20% of the maximum). As the force increased, the deformation of the box became larger, as shown in (b) and (c) of Figure 11. However, due to the holding location on the object, although the force increased, there was a smaller deformation change in parallel to the holding direction, while the defamation perpendicular to the holding direction became larger (c). This result shows that the perception of the stiffness and holding location can help the gripper to apply the right force and degree of closure.

## 4. Conclusions

In this work, an object recognition method integrating vision and haptics has been proposed. Moreover, it has been validated using an assembled robot platform with testing and comparative methods, and the results verified that the proposed method had a better performance. Despite the promising results, improvements are necessary to extend its robustness and universality due to the fact that the objects used are limited in numbers and classes. For this purpose, the addition of new features and other sensing modalities will be the main focus in the next steps. Another limitation of the proposed approach is that the type of contact during haptic exploration is point contact, which limits the data volume and reliability during sampling because only one set of data is obtained at each step and it is difficult to guarantee that contacts occur at the same location during each exploration. This can be improved by using soft tactile arrays or multiple contact tips, which will be envisioned. Moreover, in-depth research of the GCN model will be carried out. As the volume of samples becomes larger, information protection will be added, and extension methods such as injective aggregation and de-noising aggregation will be used in future work [34].

With the integration of vision and haptics, the perception of deformable objects and the localization of foreign bodies inside the object are available, which can be used to locate internal tumours inside the organ. Moreover, the control manner of manipulators can be extended to not only position (decided by visual feedback), but also force (magnitude and direction decided by haptic feedback), or both. This can be used for surgical operations and intelligent manufacturing. Taking this as a foundation, by adding other types of sensory information, the perception ability of robots can be closer to that of humans; however, this requires the accuracy of each sensing module to meet the requirements.

## Figures and Tables

**Figure 1 biomimetics-08-00086-f001:**
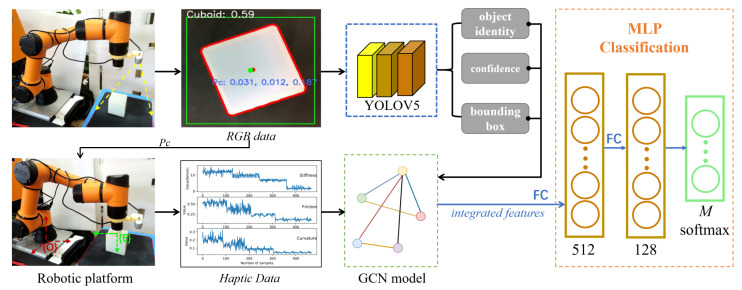
A robot platform utilising a depth camera and a force-torque sensor was assembled. With a certain field of view, the depth camera obtained the RGB data and transferred the data to the trained YOLOv5 model. The output included the confidence and bounding box for each object. On the other hand, the physical properties of the object such as stiffness, surface friction, and roughness were captured through haptic explorations. A confidence threshold $\vartheta_{r}=0.8$ was set and when the value of the obtained max confidence was greater than it, the result from the vision was trusted and no fusion was required; otherwise, the visual and haptic features would be sent to the GCN model for data integration. Finally, the fused features were sent to the connected multi-layer classifier for recognition, and the identity of the object was decided among all *M* objects.

**Figure 2 biomimetics-08-00086-f002:**
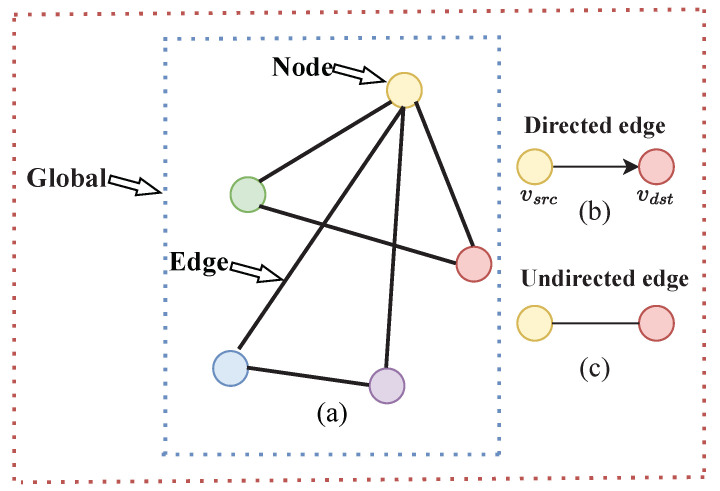
(**a**) The coloured circles represent node attributes, the lines between two circles represent edge attributes, and the whole graph represents the global attribute. In (**b**,**c**), there are directed and undirected edges to distinguish different graphs. The edges can be directed, where an edge *e* has a source node $v_{src}$ and a destination node $v_{dst}$. In this case, information flows from $v_{src}$ to $v_{dst}$. They can also be undirected, where there is no notion of source or destination nodes, and information flows in both directions.

**Figure 3 biomimetics-08-00086-f003:**
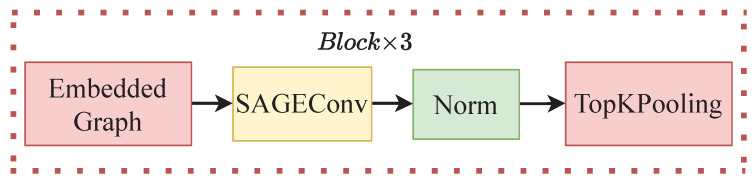
This figure shows the overview of the graph convolution model. These 4 layers form one block of the network, and in total, there are 3 blocks with the same layers.

**Figure 4 biomimetics-08-00086-f004:**
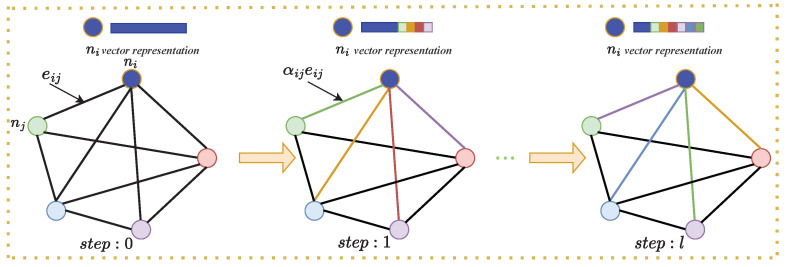
The five nodes in the figure represent the fused visual and haptic features. The line between every two nodes is used as the edge weight. Take vertex $n_{i}$ and vertex $n_{j}$ for example, at step: 0, the edge weight between these two nodes is $e_{ij}$. The vector of node $n_{i}$ has no information interaction, its feature representation is still original (in blue). At step: 1, the edge weight $e_{ij}$ is added with a new learnable normalisation parameter $\alpha_{ij}$ (the colour of the line from node $n_{i}$ to the another node represents a different normalisation parameter), node $n_{i}$ starts interacting with the other nodes and the weight parameter is updated. Its representation will change to different colours, as shown in the right figure. The cycle continues until the step: *l* (l=50) is completed.

**Figure 5 biomimetics-08-00086-f005:**
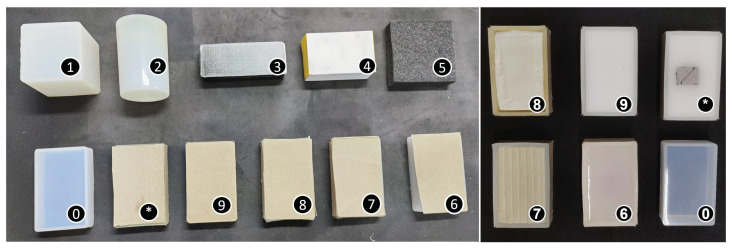
This figure shows the objects that were created for experiments. As can be seen, they are soft, and no. 1–5 are cuboid, cylinder, eraser, paper box, and black foam, respectively. While no. 0 and 6–9 are different cubes, no. 0 is the original cube, and no. 6–9 are cubes with different fillers covered with the same stiff paper. The interior fillers are silicone, plasticine, soft plasticine, and sponge. The object-* is filled with white sponge and has a metal object inside.

**Figure 6 biomimetics-08-00086-f006:**
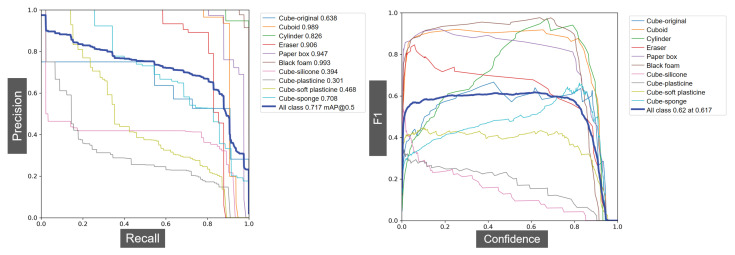
The validation results of YOLOv5 network after 500 epochs with 600 pictures. The mAP of all classes is 0.717. The accuracy and F1 score of the cubes with different fillers are relatively low (the lowest is only 0.301 and average is 0.502) due to their similar appearances. This indicates the necessity of adding haptic-related features for perception.

**Figure 7 biomimetics-08-00086-f007:**
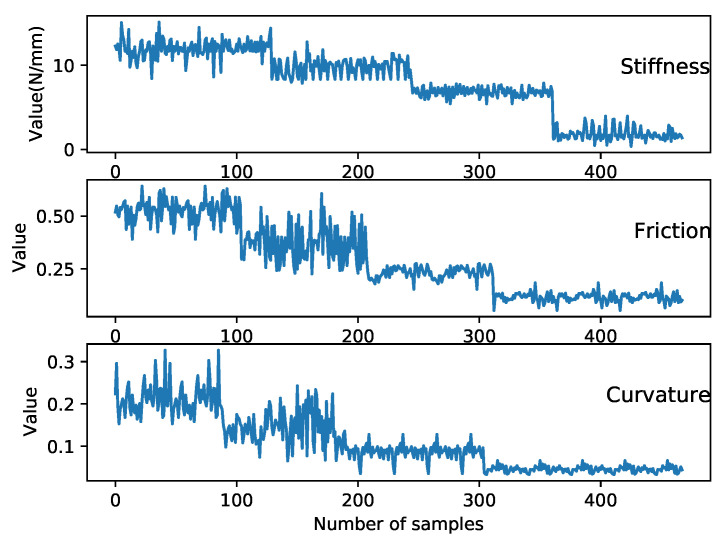
This figure shows the raw data of the haptic features extracted through the haptic explorations using the robot platform, 500 samples for 4 objects, and 125 per object. In practice, the values of the haptic features change as the object changes. For example, when the robot tries to detect the stiffness of an object, the larger the resistance to the compress force, the larger the stiffness. As shown in the figure, the obtained stiffness values for the same object is a straight line with certain vibrations due to the noise of the arm and sensor. Stepwise changes can be seen, showing the variation of the stiffness of different objects. The data samples also indicate that the haptic features are not structured and linear.

**Figure 8 biomimetics-08-00086-f008:**
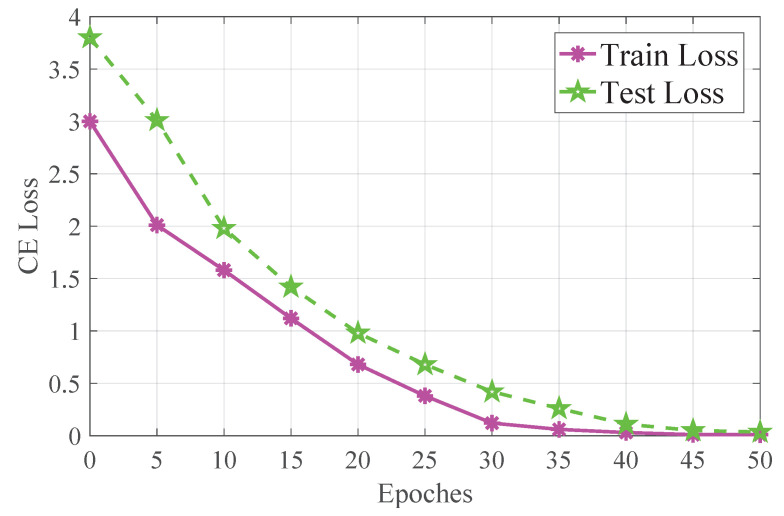
This figure shows the training result of the GCN modal. The peach line represents the train loss and the green line represents the test loss. As seen from the figure, the train loss decreases sharply in the first 25 epochs, and slowly in the last 10 epochs until the loss value reaches 0.0098 after 40 epochs. The test loss decreases sharply in the first 20 epochs, and the loss value reaches 0.13 after 45 epochs, and the loss value of the last 3 epochs increases slightly.

**Figure 9 biomimetics-08-00086-f009:**
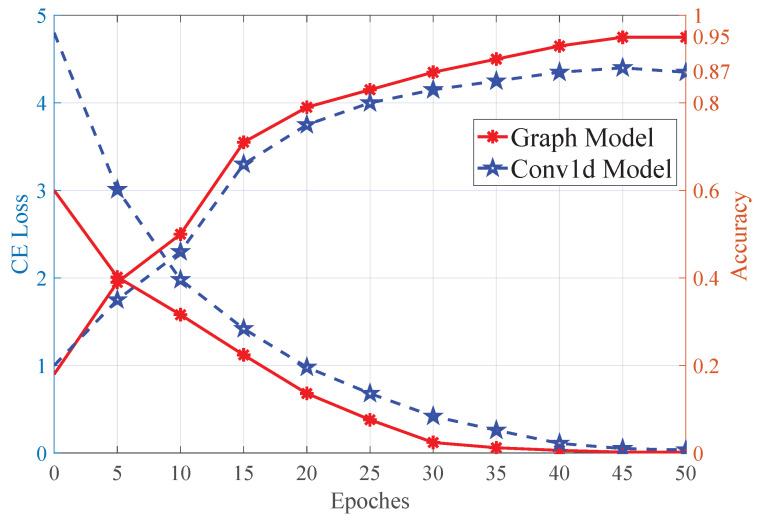
This figure shows the testing results of the GCN and Conv1d modal, with the left axis representing the training loss and the right axis showing the recognition accuracy. As seen from the figure, the train loss of the GCN modal decreases sharply in the first 25 epochs, and slowly in the last 10 epochs until the loss value reaches 0.0095 after 40 epochs. The train loss of the Conv1d model decreases sharply in the first 10 epochs, and the loss reaches 0.0518 after 45 epochs. From the comparison, it can be concluded that the GCN model has three advantages: (1) the initial loss is lower, (2) the model converges more quickly, (3) the total loss is lower.

**Figure 10 biomimetics-08-00086-f010:**
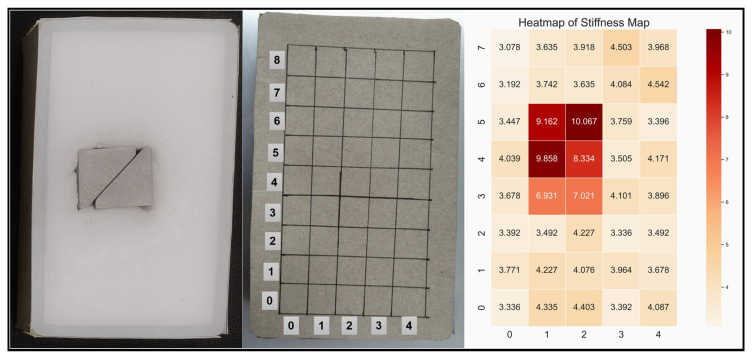
Picture on the left is the inner view of the object-*, picture in the middle shows the grids drew on its surface. The picture on the right shows the stiffness map of object-* represented with a heat-map. The values inside the picture are the stiffness at each location. During the test, the grids of 0–4 and 0–7 (vertical) were compressed. Compared with the left picture, it can be seen that the robot was able to locate the stiff object inside the sponge through haptic exploration and a vision guide.

**Figure 11 biomimetics-08-00086-f011:**
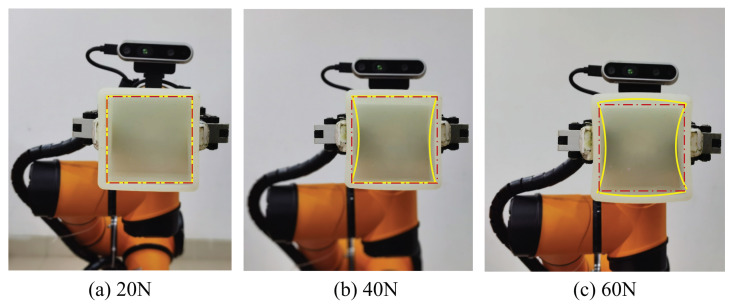
These figures show the deformation of the soft cubic box under different holding forces using the DH-Ag95 gripper. The degree of closure was fixed to 30% (28.5 mm in total and 14.25 on one side) during the test. The yellow lines represent the deformation of the box under different forces, and the red dot lines express the deformation of the box under 20% of maximum force.

**Table 1 biomimetics-08-00086-t001:** Sample example of feature values for each object.

Object I	Confidence	Bounding Box	Stiffness	Surface Friction	Roughness
0	0.85	(0.96,0.45,0.58,0.69)	-	-	-
1	0.90	(0.36,0.65,0.21,0.47)	-	-	-
2	0.87	(0.18,0.57,0.38,0.79)	-	-	-
3	0.88	(0.26,0.35,0.88,0.69)	-	-	-
4	0.95	(0.06,0.25,0.58,0.79)	-	-	-
5	0.638	(0.66,0.25,0.38,0.59)	0.753	0.255	0.091
6	0.394	(0.16,0.55,0.61,0.87)	10.014	0.394	0.108
7	0.301	(0.38,0.47,0.68,0.59)	12.059	0.052	0.034
8	0.468	(0.36,0.65,0.78,0.89)	6.817	0.114	0.056
9	0.708	(0.16,0.05,0.88,0.49)	1.859	0.034	0.052

**Table 2 biomimetics-08-00086-t002:** The recognition results when using different number of features.

Accurary	Confidence	Bounding Box	Stiffness	Surface Friction	Roughness
0.72	✓	✓	✓	-	-
0.73	✓	✓	-	✓	-
0.73	✓	✓	-	-	✓
0.80	✓	✓	✓	✓	-
0.82	✓	✓	✓	-	✓
0.83	✓	✓	-	✓	✓
0.63	✓	-	✓	✓	✓
0.62	-	✓	✓	✓	✓
0.95	✓	✓	✓	✓	✓

**Table 3 biomimetics-08-00086-t003:** Resultant recognition accuracy of different methods.

Method	Accurary
SVM	0.82
KNN	0.81
Bayesian filter	0.83
Conv1d Model	0.87
Graph Model	0.95

## Data Availability

Where data is unavailable due to patient confidentiality.

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
