# Peer review of "A Recognition Method for Soft Objects Based on the Fusion of Vision and Haptics"

_biomimetics, 2023, doi:10.3390/biomimetics8010086_

Round 1

Reviewer 1 Report

Overall, I think the paper is interesting in its proposal of using a visual-haptic fusion method for object recognition. The idea of using multiple sensing modalities to improve object recognition is promising, especially in situations where objects have similar appearances but different internal properties or materials.

The conclusion could be improved with more discussion on the limitations and potential future work. While the results seem to show that the proposed method outperforms other methods in certain cases, it would be helpful to have more information on how the method performs in other scenarios and how it compares to other fusion methods. Additionally, a discussion on the potential applications and real-world implications of the proposed method would also be useful in understanding the potential impact of the work.

Typos:

* Line 352: "Defamation" should be "Deformation"

Reviewer 2 Report

This paper presents a model of system to solve a problem of verification models by using graph neural networks. Here are comments:

1.      How is model new in case of information protection? If scalable system is changing how information is to be protected. What type of security aspects are applied in this model to be more advanced from previous versions? Make better explanations of security in your model.

2.      Related models to extend: Improving performance and efficiency of Graph Neural Networks by injective aggregation, De-Noising Aggregation of Graph Neural Networks by Using Principal Component Analysis.

3.      It is not clear how you transform information into nodes. Which features are considered to be node?

4.      How is information indexed in your model? What type of indexation is used? Can you compere different models of indexation? How each of these models influence performance of your system? Make experiments and discuss results.

5.      What are limitations of your model? What is max and what is optimum number of nodes in this model? Make detailed discussion and show each case with results.

6.      There are no tests of the model depending on the data amount in the system. How the cost of operation goes with different data on the network in terms of efficiency? Make tests and show them.

7.      There are no comparisons to other models and no tests on different scenarios in your proposed network. Make better examinations and discuss these results to show your advances.

Round 2

Reviewer 2 Report

ok